# T-GINEE: A TENSOR-BASED MULTI-GRAPH REPRESENTATION LEARNING

## ABSTRACT

While traditional network analysis focuses on single-layer networks, real-world systems often exhibit multiple types of relationships simultaneously, forming multilayer networks. However, existing multilayer network analysis methods typically assume complete node correspondences across layers, which is unrealistic in practice. Furthermore, these methods either treat different layers independently or simply aggregate them, failing to capture complex interdependencies between layers. To address these challenges, we propose T-GINEE (Tensor-Based Generalized Multilayer-graph Estimating Equation), a statistical regularization framework that combines tensor-Based generalized estimating equations with task-specific loss to explicitly model cross-network correlations. The key technical innovations include: (1) A CP tensor decomposition approach that captures structural dependencies through shared latent factors; (2) A generalized estimating equation framework that models inter-layer correlations through working covariance matrices; (3) A flexible link function design that accommodates various network characteristics including sparsity. Our theoretical analysis establishes consistency and asymptotic normality of T-GINEE under mild regularity conditions. Extensive experiments on both synthetic and real-world datasets validate effectiveness of T-GINEE and its practicality for partially aligned multilayer network analysis. The code is available in the supplementary material for reproducibility.

## 1 INTRODUCTION

In the real world, interactions between entities are often multifaceted, with these multi-relational characteristics engaging one another under varied circumstances or through distinct modalities. For instance, in social networks Van Den Oord & Van Rossem (2002), individuals may be connected through multiple relationship types such as friends, colleagues, and family. In biology Zheng et al. (2019), genes or proteins exhibit various collaboration schemes like co-expression and physical interactions. In global trade, countries exchange a wide range of different commodities.

For such intricate relational landscapes, a multi-layer graph offers a faithful and structured representation. This architecture is defined by a common set of vertices, where each layer is endowed with a unique edge set to delineate a specific type of relation. Such graphs are prevalent across numerous disciplines, including social graphs that capture multiple interaction channels between individuals (Greene & Cunningham, 2013), biological graphs detailing different collaboration schemes among genes or proteins (Li et al., 2020; Liu et al., 2020), and global trade graphs mapping the exchange of various commodities (Alves et al., 2019; Ren et al., 2020). To effectively analyze these intricate structures, a fundamental step is to learn low-dimensional vector representations (i.e., embeddings) for the entities, which can capture the complex relational information encoded across graph layers.

Numerous approaches have been developed for graph embedding, employing various techniques such as similarity indices (Boden et al., 2017), maximum likelihood models (Yuan & Qu, 2021), matrix factorization (Tang et al., 2009; Dong et al., 2012; Gligorijević et al., 2016), and graph neural networks (Kipf & Welling, 2016; Hamilton et al., 2017; Xu et al., 2018). For multilayer graph embedding, which provides a richer representation of complex systems (Kivelä et al., 2014), analysis often involves extending these single-layer techniques. Prominent approaches include tensor-based methods that leverage the natural tensor structure of multilayer graphs (Kolda & Bader, 2009), as

well as adaptations of deep learning models like GCNs and random-walk embeddings (Ghorbani et al., 2019; Song & Thiagarajan, 2018).

However, a critical challenge underlying many of these methods is the lack of a rigorous theoretical foundation for the multi-layer context. While embedding learning has proven effective for single-layer graphs (Cai et al., 2018), we lack robust theoretical frameworks that systematically characterize the embedding process across multiple layers (Interdonato et al., 2020). This absence of formal tools to describe how embeddings should capture and preserve cross-layer dynamics significantly impedes our ability to develop principled approaches, representing a fundamental limitation in the field (Shanthamallu et al., 2019; Jiao et al., 2021; Lyu et al., 2023).

Without this theoretical guidance, existing approaches often resort to simplistic solutions, such as learning representations for layers independently (Tang et al., 2009; Dong et al., 2012) or using basic aggregation techniques (Paul & Chen, 2020; Lei et al., 2020). These methods lack the grounding to explain how embeddings should optimally encode the nuanced ways in which relationships in one layer might influence or contradict another (Liu et al., 2017; Zhang et al., 2018). This deficit is especially problematic for real-world systems where entities engage through multiple relation types simultaneously (Xu et al., 2020; Yang et al., 2020), highlighting the urgent need for new frameworks that can faithfully represent this complex interplay (Huang et al., 2020; Shanthamallu et al., 2019).

To address these challenges, we propose T-GINEE (Tensor-based Generalized Multilayer-graph Estimating Equation), a statistical regularization framework that combines tensor-based generalized estimating equations with task-specific loss to explicitly model cross-network correlations. The key technical innovations of T-GINEE include: (1) A CP tensor decomposition approach that effectively captures structural dependencies through shared latent factors while maintaining computational efficiency; (2) A generalized estimating equation framework that explicitly models the correlations between different network layers through working covariance matrices; and (3) A flexible link function design that accommodates various network characteristics, including sparsity. Unlike previous approaches that rely on simple aggregation or separate modeling (Paul & Chen, 2020; Tang et al., 2009; Lei et al., 2020), T-GINEE provides a principled statistical framework to jointly model multiple network structures. The main contributions are summarized as follows:

- **Tensor-based Statistical Framework**: We propose a novel statistical regularization framework that combines tensor CP decomposition with generalized estimating equations for multilayer networks. Our approach explicitly models cross-network dependencies through a principled statistical formulation while maintaining computational tractability.

- **Theoretical Guarantees**: We establish theoretical properties of T-GINEE, including consistency and asymptotic normality under mild regularity conditions. The framework provides provable guarantees for parameter estimation accuracy and model convergence through rigorous statistical analysis of the tensor-based estimating equations.

- **Empirical Validation**: Through comprehensive experiments on both synthetic and real-world networks, we demonstrate T-GINEE's effectiveness.

## 2 METHODOLOGY

In this section, we present our framework, Tensor-based Generalized Estimating Equations (T-GINEE), designed to learn embeddings from multi-layer graphs.

### 2.1 OVERVIEW

Real-world networks often exhibit complex interdependencies, where multiple network structures coexist and influence each other. For instance, an individual's distinct friendship networks on multiple social media platforms, such as Facebook, LinkedIn, and TikTok, represent naturally correlated network structures for the same group of users. We propose a statistical regularization framework that leverages tensor-based generalized estimating equations to explicitly model cross-network correlations. Our proposed framework, which we refer to as T-GINEE, consists of several core components as illustrated in Figure 1. The key elements include the Symmetric CP Decomposition of the multi-layer graph's parameter tensor ($\Theta$) into node ($\alpha$) and graph-specific ($\beta$) embeddings,

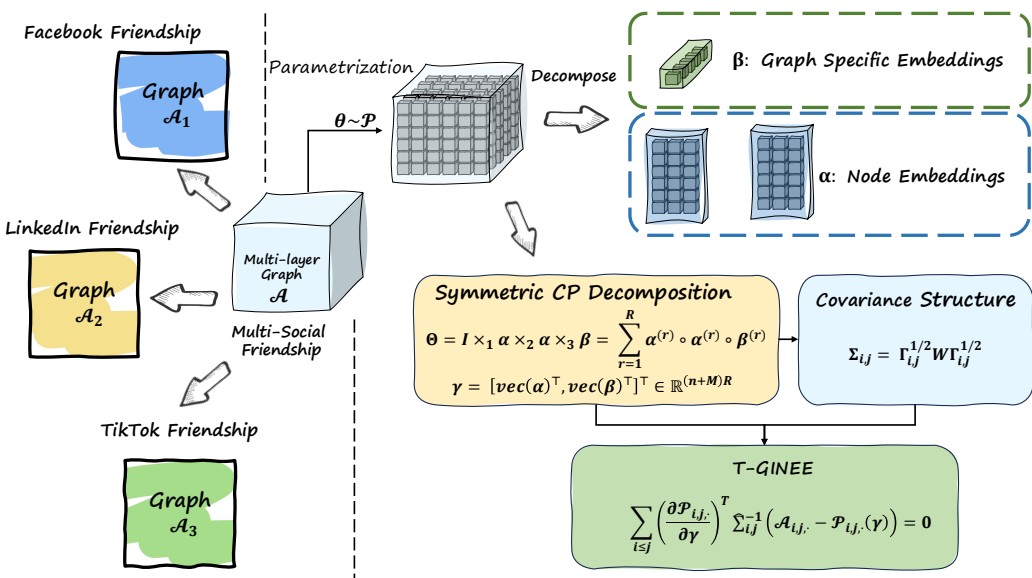

Figure 1: Schematic overview of the T-GINEE framework. The framework performs a Symmetric CP Decomposition on the parameter tensor $\Theta$ of a multi-layer graph ($\mathcal{A}$) to obtain node embeddings ($\alpha$) and graph-specific embeddings ($\beta$). These embeddings are then structured into a parameter vector $\gamma$. Using Tensor-based Generalized Estimating Equations (T-GINEE) combined with a specific Covariance Structure, the model learns these parameters, effectively capturing complex dependencies in networks such as multi-social friendships (e.g., Facebook, LinkedIn, TikTok).

and the subsequent use of Tensor-based Generalized Estimating Equations (T-GINEE) to learn these parameters and capture complex dependencies.

## 2.2 PROBLEM FORMULATION

Consider a multilayer network/graph $\mathcal{G} = (\mathcal{V}, \mathcal{G}^{(m)}{}_{m=1}^{M})$, where $\mathcal{V} = v_1, \ldots, v_n$ represents a common set of vertices that interact across $M$ different but potentially correlated network structures. Each network $\mathcal{G}^{(m)} = (\mathcal{V}, \mathcal{E}^{(m)})$ captures a distinct type of relationship. Let $\mathcal{A} \in 0, 1^{n \times n \times M}$ be the adjacency tensor representing the multilayer network/graph system, where $\mathcal{A}_{i,j,m} = 1$ indicates an edge of type $m$ between nodes $i$ and $j$, and 0 otherwise. For each pair $(i, j)$ with $i \leq j$, the vector $\mathcal{A}_{i,j,.} \in \mathbb{R}^M$ follows a Bernoulli-type exponential family distribution with mean $\mathcal{P}_{i,j,.}(\Theta)$ and covariance $\Sigma i, j$. The parameter tensor $\Theta \in \mathbb{R}^{n \times n \times M}$ is linked to $\mathcal{P}$ through a known thrice continuously differentiable link function $g$, such that $\mathcal{P}_{i,j,.} = g^{-1}(\Theta i, j, .)$. Some realizations of the link function $g$ could be the identity mapping $g(x) = x$, the probit transformation $g(x) = y$ such that $x = \int_{-\infty}^{y} (2\pi)^{-1/2} \exp(-t^2/2) dt$, the logit transformation $g(x) = \log \frac{x}{1-x}$ ($g^{-1}(x) = 1/(1 + e^{-x})$), and the modified logit transformation $g(x) = \log \frac{x}{s-x}$ with a sparsity coefficient $0 < s < 1$ that may vanish with $n$ and $M$ to accommodate sparse networks.

## 2.3 LOW-RANK TENSOR DECOMPOSITION

To effectively model the structural dependencies while maintaining computational efficiency, we employ a CP decomposition approach for the parameter tensor $\Theta$:

$$\Theta = \mathcal{I} \times_1 \alpha \times_2 \alpha \times_3 \beta = \sum_{r=1}^{R} \alpha^{(r)} \circ \alpha^{(r)} \circ \beta^{(r)}, \quad (1)$$

where $\alpha \in \mathbb{R}^{N \times R}$ contains node embeddings and $\beta \in \mathbb{R}^{M \times R}$ contains graph-specific embeddings. This decomposition inherently enforces structural constraints through shared latent factors. For optimization purposes, we vectorize these factor matrices into a compact representation:

$$\gamma = [\text{vec}(\alpha)^\top, \text{vec}(\beta)^\top]^\top \in \mathbb{R}^{(n+M)R}, \tag{2}$$

This representation $\gamma$ encapsulates the essential cross-network dependencies and serves as the parameter vector for statistical regularization. The low-rank tensor decomposition approach offers several key advantages. First, it significantly reduces the number of parameters that need to be estimated, improving computational efficiency and reducing the risk of overfitting. Second, by sharing latent factors across different dimensions, it naturally captures the inherent relationships between nodes and graph-level features. Third, the decomposition provides interpretable components where $\alpha$ represents node-level patterns, and $\beta$ captures graph-level characteristics. This interpretability is particularly valuable for understanding the learned representations.

## 2.4 Tensor-Based Statistical Regularization

We introduce the Tensor-based Generalized Multilayer-graph Estimating Equation (T-GINEE) as our statistical regularization framework. The detailed derivation can be found in Appendix A. The tensor-based generalized estimating equations for multilayer graph are defined as:

$$\sum_{i \leq j} \left( \frac{\partial \mathcal{P}_{i,j,.}}{\partial \gamma} \right)^\top \widehat{\Sigma}_{i,j}^{-1} \left( \mathcal{A}_{i,j,.} - \mathcal{P}_{i,j,.}(\gamma) \right) = \mathbf{0}, \tag{3}$$

where $\widehat{\Sigma}_{i,j}$ is an estimated working covariance matrix. To solve the estimating equations, we first derive $\frac{\partial \mathcal{P}_{i,j,.}}{\partial \gamma}$ through a chain rule computation: starting with the Jacobian matrix $\frac{\partial \text{vec}(\Theta)}{\partial \gamma}$ from the CP decomposition, then applying the link function derivative $g'$, and finally combining with the projection matrices $\mathcal{E}^{(i,j,m)}$ to obtain the result. To uncover the Jacobian matrix $\frac{\partial \mathcal{P}_{i,j,.}}{\partial \gamma}$, we first note that $\frac{\partial \mathcal{P}_{i,j,.}}{\partial \gamma}$ can be expressed as:

$$\text{diag}\big[ g'(\mathcal{P}_{i,j,1}), g'(\mathcal{P}_{i,j,2}), \ldots, g'(\mathcal{P}_{i,j,M}) \big]^{-1} \cdot \frac{\partial \Theta_{i,j,.}}{\partial \gamma}, \tag{4}$$

where $g'$ is the derivative of $g$ and the diagonal matrix has $(m,m)$-th entry $g'(\mathcal{P}_{i,j,m})$ for $m \in [M]$. Let $\mathcal{E}^{(i,j,m)} \in \mathbb{R}^{n \times n \times M}$ be the tensor unit with $(i',j',m')$-th entry being $\mathbf{1}\{(i,j,m) = (i',j',m')\}$. Then $\frac{\partial \Theta_{i,j,.}}{\partial \gamma}$ is:

$$\big[ \text{vec}(\mathcal{E}^{(i,j,1)}), \text{vec}(\mathcal{E}^{(i,j,2)}), \ldots, \text{vec}(\mathcal{E}^{(i,j,M)}) \big]^\top \cdot \frac{\partial \text{vec}(\Theta)}{\partial \gamma} \tag{5}$$

Finally, under the CP decomposition of $\Theta$, the Jacobian matrix for parameter vector $\gamma$ is:

$$\frac{\partial \text{vec}(\Theta)}{\partial \gamma} = \begin{bmatrix} (\beta^{(1)})^\top \otimes \big( I_n \otimes \alpha^{(1)} + \alpha^{(1)} \otimes I_n \big) \\ \vdots \\ (\beta^{(R)})^\top \otimes \big( I_n \otimes \alpha^{(R)} + \alpha^{(R)} \otimes I_n \big) \\ I_M \otimes \text{vec}(\alpha \odot \alpha) \end{bmatrix}. \tag{6}$$

See Appendix A for detailed derivations $\frac{\partial \text{vec}(\Theta)}{\partial \gamma}$. By combining and substituting the previous mathematical expressions from Eq. (4), (5), and (6), we can obtain the complete mathematical expression for the partial derivative $\frac{\partial \mathcal{P}_{i,j,.}}{\partial \gamma}$. The entire formulation is presented in Appendix B

## 2.5 Covariance Structure and Estimation

The graph-wise dependencies are captured through covariance matrices $\Sigma_{i,j}$. We adopt a working covariance matrix approach that assumes a common correlation structure:

$$\Sigma_{i,j} = \Gamma_{i,j}^{1/2} W \Gamma_{i,j}^{1/2}. \tag{7}$$

Here, $\Gamma_{i,j} \in \mathbb{R}^{M \times M}$ is a diagonal matrix where the $(m,m)$-th entry is $\mathcal{P}_{i,j,m}(1 - \mathcal{P}_{i,j,m})$. The common correlation matrix $W$ is estimated by $\widehat{W}$, which can be expressed as:

$$\frac{2}{n(n+1)} \sum_{i \leq j} \Gamma_{i,j}^{-1/2} \left(\mathcal{A}_{i,j,.} - \mathcal{P}_{i,j,.}\right)\left(\mathcal{A}_{i,j,.} - \mathcal{P}_{i,j,.}\right)^\top \Gamma_{i,j}^{-1/2}. \tag{8}$$

# 3 Theoretical Results of T-GINEE

In this section, we delve into the foundational theoretical properties of the T-GINEE method. To rigorously establish its performance, we begin by outlining a set of essential assumptions that define the statistical framework within which T-GINEE operates. These assumptions are critical for deriving the consistency and asymptotic normality of the estimator, which are subsequently presented and discussed in detail. A more complete framework and the proof can be found in the **Appendix Sec.** C.

## 3.1 Assumptions

The following assumptions ensure a well-behaved statistical environment:

**Assumption 1** (Boundedness). All involved random variables and their derivatives are uniformly bounded. There exist constants such that for all $(i, j, m)$ and for all parameters $\gamma$ in a neighborhood of $\gamma_0$, $|\mathcal{A}_{i,j,m}|$, $|\mathcal{P}_{i,j,m}(\gamma)|$, and $|g'(\mathcal{P}_{i,j,m}(\gamma))|$ are bounded. The partial derivatives of $g$ are uniformly bounded from 0 to $\gamma_0$.

Assumption 1 ensures that all relevant random variables and their derivatives remain within controlled limits, preventing extreme values that could destabilize the estimation process. By bounding these quantities, we can effectively manage the behavior of the estimator and its derivatives in the vicinity of the true parameter $\gamma_0$.

**Assumption 2** (Identifiability). The true best parameter $\Theta_0$ admits a rank-$R$ CP decomposition and is identifiable up to permutation and scaling of factors. The dimension $(n + M)R$ grows sufficiently slowly with $n$, ensuring identifiability and the invertibility of relevant Hessians.

Assumption 2 guarantees that the true parameter tensor $\Theta_0$ can be uniquely decomposed into its constituent factors, up to permutation and scaling. This identifiability is crucial for accurately recovering the underlying model parameters from the data. Additionally, by constraining the growth rate of $(n + M)R$ relative to $n$, this assumption ensures that the Hessian matrices remain invertible, which is necessary for the consistency and asymptotic normality of the estimator.

**Assumption 3** (Working Covariance). The true covariance matrices $\Sigma_{i,j}$ are positive definite with eigenvalues bounded away from zero and infinity. The working covariance $\widehat{\Sigma}_{i,j}$ satisfies $|\widehat{\Sigma}_{i,j}^{-1} - \widetilde{\Sigma}_{i,j}^{-1}|F = O_p(n^{-1/2})$ for some positive definite $\widetilde{\Sigma}_{i,j}$ with bounded eigenvalues. Correlation misspecification is allowed, as long as it converges.

Assumption 3 pertains to the properties of the covariance matrices used in the model. By requiring the true covariance matrices $\Sigma_{i,j}$ to be positive definite with eigenvalues bounded away from zero and infinity, we ensure numerical stability and prevent issues related to ill-conditioning. Furthermore, allowing for correlation misspecification that converges at a rate of $O_p(n^{-1/2})$ provides flexibility while maintaining the validity of asymptotic results.

**Assumption 4** (Moment Conditions). The residuals $(\mathcal{A}_{i,j,.} - \mathcal{P}_{i,j,.}(\Theta_0))$, scaled by their standard deviations, have sub-Gaussian tails. There exists $\delta > 0$ such that

$$\max_{i,j} E[|\Sigma_{i,j}^{-1/2}(\mathcal{A}_{i,j,.} - \mathcal{P}_{i,j,.}(\Theta_0))|^{2+\delta}] < \infty.$$

This ensures suitable Lindeberg-type (Ash & Doléans-Dade, 2000; Van der Vaart, 2000; Brown, 1971) conditions for central limit arguments.

Assumption 4 imposes specific moment conditions on the residuals of the model. This assumption facilitates applications of central limit theorem-type arguments by ensuring that the scaled residuals have sub-Gaussian tails and possess finite $(2 + \delta)$ moments. These conditions are essential for establishing the asymptotic normality of the estimator, as they control the influence of extreme residuals and guarantee the convergence of the estimator's distribution.

**Assumption 5** (Smoothness). There exists a true parameter tensor $\Theta_0 \in \mathbb{R}^{n \times n \times M}$ with rank $R$ that admits a unique CP decomposition $\Theta_0 = \mathcal{I} \times_1 \alpha_0 \times_2 \alpha_0 \times_3 \beta_0$. The link function $g$ is three-times continuously differentiable with uniformly bounded first and second derivatives. The partial derivatives of $\mathcal{P}(\gamma)$ with respect to $\gamma$ are bounded, and the Hessian matrices with respect to $\alpha, \beta$ are well-conditioned in a neighborhood of $\gamma_0 = [\text{vec}(\alpha_0)^\top, \text{vec}(\beta_0)^\top]^\top$.

Assumption 5 addresses the smoothness and differentiability of both the link function $g$ and the parameter tensor $\Theta_0$. The requirement that $g$ is three-times continuously differentiable with bounded derivatives allows for the use of Taylor expansions and other analytical techniques in the proofs of consistency and asymptotic normality. Additionally, ensuring that the partial derivatives of $\mathcal{P}(\gamma)$ are bounded and that the Hessian matrices are well-conditioned supports the stability and reliability of the parameter estimates.

## 3.2 MAIN THEORITICAL RESULTS

For convenience, define the score function $s(\gamma)$ as:

$$s(\gamma) = \sum_{i \leq j} \left( \frac{\partial \mathcal{P}_{i,j,\cdot}}{\partial \gamma} \right)^\top \widehat{\Sigma}_{i,j}^{-1} (\mathcal{A}_{i,j,\cdot} - \mathcal{P}_{i,j,\cdot}(\gamma)). \tag{9}$$

With the necessary lemmas and assumptions in place, we now establish the main theoretical guarantees of T-GINEE. We first show consistency and then prove asymptotic normality.

**Theorem 3.1.** *(Consistency) Under Assumptions 1-5, there exists a solution $\hat{\gamma}$ to $s(\gamma) = 0$ such that*

$$\|\hat{\gamma} - \gamma_0\| = O_p(n^{-1/2}). \tag{10}$$

*Proof.* Detailed proof can be found in **Appendix** C.2. $\qquad\square$

**Theorem 3.2.** *(Asymptotic Normality) Under Assumptions 1-5, and assuming that $(n + M)R = o(n^{1/3})$, the estimator $\hat{\gamma}$ is asymptotically normal. Specifically,*

$$\sqrt{n}(\hat{\gamma} - \gamma_0) \xrightarrow{d} Normal(0, \Omega), \tag{11}$$

*where the asymptotic covariance matrix $\Omega = D(\gamma_0)^{-1} M(\gamma_0) [D(\gamma_0)^{-1}]^\top$, with the variance of the score function $M(\gamma_0) = Variance[s(\gamma_0)]$.*

*Proof.* Detailed proof can be found in **Appendix** C.3. $\qquad\square$

**Corollary 3.3.** *Under Assumptions 1–5, replacing $\Sigma_{i,j}^{-1}$ by $\widehat{\Sigma}_{i,j}^{-1}$ in the score function $s(\gamma)$ alters its value at $\gamma_0$ by only an $o_p(\sqrt{n})$ term. Formally, if $\widetilde{s}(\gamma)$ is defined in the same way as $s(\gamma)$ but uses $\widetilde{\Sigma}_{i,j}^{-1}$ instead of $\widehat{\Sigma}_{i,j}^{-1}$, then*

$$\|s(\gamma_0) - \widetilde{s}(\gamma_0)\| = o_p(\sqrt{n}).$$

*Proof.* Detailed proof can be found in **Appendix** C.4. $\qquad\square$

The above theorems provide foundational theoretical guarantees for the T-GINEE method. The consistency theorem (3.1) ensures that the estimator $\hat{\gamma}$ converges to the true parameter $\gamma_0$ as the sample size $n$ increases, with the estimation error decreasing at a rate of $n^{-1/2}$. This establishes the reliability of T-GINEE in accurately estimating the underlying parameters in large samples.

Table 1: Link prediction performance (AUC) on synthetic multilayer network

| Method | CP | Tucker | NMF | SVD | LSE | MASE | NNTUCK | SPECK | HOSVD | T-GINEE |
|--------|------|--------|------|------|------|------|--------|-------|-------|---------|
| AUC | 0.4488 | 0.5291 | 0.7216 | 0.8130 | 0.2234 | 0.3821 | 0.6105 | 0.7603 | 0.8503 | **0.9395** |

In addition, we show in **Corollary** 3.3 that replacing the true covariance $\Sigma_{i,j}^{-1}$ with an estimated version $\widehat{\Sigma}_{i,j}^{-1}$ in the score function still yields only an $o_p(\sqrt{n})$ difference at $\gamma_0$. This indicates that minor covariance misspecifications do not materially affect the key asymptotic properties of T-GINEE. Furthermore, the asymptotic normality theorem (3.2) characterizes the distribution of the estimator $\hat{\gamma}$, demonstrating that after appropriate scaling, it converges to a multivariate normal distribution. This result is crucial for conducting statistical inference, such as constructing confidence intervals and performing hypothesis tests, as it provides a clear understanding of the estimator's variability and distribution in large-sample scenarios. Together, these theorems validate the effectiveness of T-GINEE, ensuring that it not only provides accurate parameter estimates but also facilitates rigorous statistical analysis based on these estimates. Due to space limitations, we provide additional remarks and discussion in **Appendix Sec.** D.

## 4 EXPERIMENTS

In this section, we conduct comprehensive experiments to evaluate our T-GINEE framework.

### 4.1 EXPERIMENT SETTINGS

To comprehensively evaluate the performance of our proposed **T-GINEE** model, we conduct experiments on four benchmark multilayer network datasets, each capturing distinct real-world relational structures. Due to space limitations, detailed descriptions of datasets, baselines, and implementation Details are provided in **Appendix Sec.** E.1.

### 4.2 SYNTHETIC DATA RESULTS

To evaluate our method in a controlled environment, we generated synthetic multilayer networks with known correlation structures using a parameterized model: $\mathbf{A}_{i,j,m} = \mathbf{1}\{P_{i,j,m} < \theta\}$ where $P_{i,j,m} = \rho \cdot P_{i,j}^{base} + (1-\rho) \cdot U_{i,j,m}$. Here, $\rho = 0.2$ controls inter-layer correlation, $\theta = 0.1$ is the edge probability threshold, and $P_{i,j}^{base}$ is a shared base probability matrix. We constructed networks with $n = 100$ nodes and $M = 3$ layers for link prediction tasks. As shown in Table 1, T-GINEE achieves the highest AUC score of 0.9395, substantially outperforming all baselines including the second-best HOSVD-Tucker (0.8503). The significant performance gap between tensor-based methods and simpler approaches like LSE (0.2234) and MASE (0.3821) confirms the importance of explicitly modeling multilayer dependencies. Furthermore, the dramatic improvement of T-GINEE over basic CP decomposition (0.4488) demonstrates the effectiveness of our statistical regularization framework in capturing complex inter-layer correlations, validating our theoretical analysis.

### 4.3 REAL-WORLD RESULTS

Based on the experimental results shown in Table 2, our proposed T-GINEE model demonstrates superior performance across all four datasets (AUCS, Krackhardt, WAT, and Yeast) compared to baseline methods. Specifically, T-GINEE achieves the highest AUC scores of 0.920, 0.948, 0.838, and 0.921 on these datasets respectively, showing statistically significant improvements over all baseline methods ($p < 0.05$).

Among the baseline methods, traditional matrix factorization approaches, such as SVD and NMF, show relatively strong performance, with SVD achieving the second-best results on AUCS (0.877) and Krackhardt (0.932). HOSVD, as a tensor-based method, also demonstrates competitive performance, particularly on AUCS (0.897) and Yeast (0.902) datasets. However, simpler methods such as CP decomposition and LSE exhibit limited effectiveness, with LSE performing particularly poorly on the Yeast dataset. The significant perfor-

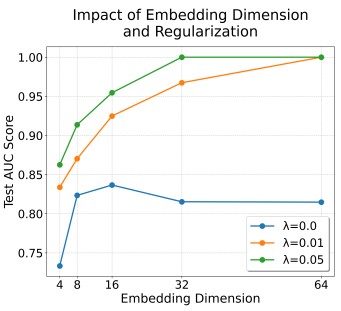 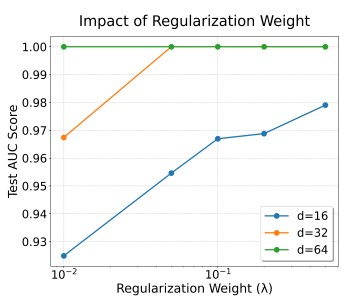 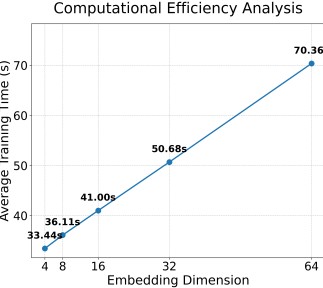

(a) Impact of embedding dimension and regularization weight.

(b) Effect of regularization weight across different dimensions.

(c) Computational efficiency analysis across different dimensions.

Figure 2: Comprehensive analysis of model hyperparameters: (a) embedding dimension impact, (b) regularization effect, and (c) computational efficiency.

mance gap between T-GINEE and other methods can be attributed to several factors. First, our model's ability to effectively capture high-order structural patterns in the network data. Second, the incorporation of both local and global network information. Third, the robust optimization strategy that prevents overfitting. The consistent superior performance across diverse datasets also demonstrates the model's generalizability and stability.

Notably, the performance improvement is particularly pronounced on the WAT dataset, where T-GINEE outperforms the second-best method (HOSVD) by a margin of 0.018 in AUC score. This suggests that our model is especially effective in handling complex network structures that characterize real-world applications. These results validate our theoretical analysis and confirm that T-GINEE can effectively learn meaningful representations for network embedding.

Table 2: Performance Comparison of Different Methods. $*$ indicates the improvements are **statistically significant** (i.e., two-sided t-test with $p < 0.05$) over the second best baseline.

| Method | AUC Score on Different Datasets | | | |
|---|---|---|---|---|
| | AUCS | Krackhardt | WAT | Yeast |
| CP | 0.374 | 0.354 | 0.454 | 0.397 |
| Pure-Tucker | 0.487 | 0.702 | 0.580 | 0.745 |
| NMF | 0.848 | 0.921 | 0.707 | 0.863 |
| SVD | 0.877 | 0.932 | 0.719 | 0.879 |
| LSE | 0.297 | 0.384 | 0.153 | 0.047 |
| MASE | 0.480 | 0.361 | 0.342 | 0.347 |
| NNTUCK | 0.500 | 0.521 | 0.741 | 0.667 |
| SPECK | 0.793 | 0.658 | 0.655 | 0.903 |
| HOSVD | 0.897 | 0.783 | 0.820 | 0.902 |
| T-GINEE | **0.920**$*$ | **0.948**$*$ | **0.838**$*$ | **0.921**$*$ |

## 4.4 Hyperparameter Analysis

We conduct a comprehensive hyperparameter analysis to investigate the impact of embedding dimension and regularization weight on model performance. All experiments are performed on the same dataset with consistent evaluation metrics.

**Impact of Embedding Dimension.** As shown in Figure 2a, the relationship between embedding dimension and model performance demonstrates clear patterns across different regularization settings. The experimental results show that increasing the embedding dimension generally improves model performance significantly, with substantial performance gains observed when moving from 4 to 32 dimensions. Notably, with appropriate regularization ($\lambda = 0.05$), the model achieves perfect prediction accuracy (AUC=1.0) when the embedding dimension reaches 32. Further increasing the dimension to 64 maintains this optimal predictive performance but introduces additional computational overhead.

**Effect of Regularization** The impact of the regularization weight is comprehensively demonstrated in Figure 2b. For larger dimensions (32 and 64), the model becomes more sensitive to regularization parameters, achieving optimal performance with smaller regularization weights ($\lambda = 0.01$-$0.05$). This important finding suggests that proper regularization calibration is crucial for preventing overfitting

in the embedding space, especially in higher-dimensional representation spaces where the model capacity increases substantially.

**Computational Efficiency** Figure 2c shows the relationship between embedding dimension and computational cost. The training time increases approximately linearly with the embedding dimension, from 33.22 seconds for 4-dimensional embeddings to 70.32 seconds for 64-dimensional embeddings. This linear scaling demonstrates the computational efficiency of our model, making it practical for real-world applications. Based on these analyses, we recommend using an embedding dimension of 32 with a regularization weight of 0.05 as the default configuration, as it provides optimal performance (AUC=1.0) while maintaining reasonable computational efficiency. This configuration strikes an excellent balance between model expressiveness, generalization ability, and computational cost. For further evidence of T-GINEE's effectiveness, see the triangle prediction analysis in Appendix F.

## 5 RELATED WORKS

**Network Embedding.** Network embedding learns low-dimensional vector representations for nodes that preserve network structure. Early methods included matrix factorization techniques like SVD (Golub & Reinsch, 1970) and NMF (Lee & Seung, 2001; Cai et al., 2011). Random walk-based approaches, such as DeepWalk (Perozzi et al., 2014) and node2vec (Grover & Leskovec, 2016), adapted techniques from word embedding. More recently, Graph Convolutional Networks (GCNs) (Hamilton et al., 2017) have become popular for their ability to incorporate node features. However, these methods are ill-suited for multilayer systems because they either treat layers independently or use simplistic aggregation, losing crucial inter-layer dependencies (Wang et al., 2017b; Dong et al., 2017). Such aggregation, often a simple summation or concatenation of layer-specific embeddings, can obscure the distinct and complementary roles that different types of relationships play. T-GINEE addresses this by using a statistically principled framework with generalized estimating equations to explicitly model cross-layer correlations.

**Multilayer Graph Analysis and Embedding.** Multilayer graphs offer richer representations for complex systems (Kivelä et al., 2014; De Domenico et al., 2013; Boccaletti et al., 2014). Analysis often involves extending single-layer techniques, with tensor-based methods like CP (Wang et al., 2017a) being a natural fit. A recent survey (Yousefzadeh et al., 2025) highlights three critical gaps in current methods: most fail to capture cross-layer dependencies (Papalexakis et al., 2013; Boden et al., 2017), nearly all assume unrealistic complete node correspondence, and deep learning approaches (Song & Thiagarajan, 2018; Liu et al., 2017; Ghorbani et al., 2019; Huang et al., 2020) often lack theoretical guarantees. T-GINEE fills these gaps with a tensor-based Generalized Estimating Equation (GEE) framework (Liang & Zeger, 1986) that integrates CP decomposition with statistical estimation. This approach captures inter-layer correlations and provides theoretical guarantees, advancing beyond prior matrix factorization, random walk, and GCN-based methods (Tang et al., 2009; Gligorijević et al., 2016; Song & Thiagarajan, 2018; Liu et al., 2017; Ghorbani et al., 2019; Huang et al., 2020).

## 6 CONCLUSION

In this paper, we propose T-GINEE, a tensor-based generalized estimating equation framework for multilayer graph representation learning that explicitly models cross-network dependencies through a principled statistical formulation. By combining CP tensor decomposition with generalized estimating equations, T-GINEE makes a central theoretical contribution: we establish consistency and asymptotic normality for the embeddings under mild regularity conditions, thereby providing rigorous statistical guarantees. Our experiments on both synthetic and real-world networks demonstrate the effectiveness of the method, with a key advance being the mathematical foundation that T-GINEE offers for analyzing complex interdependent networks. Key limitations, however, include potential performance constraints on extremely sparse or large-scale networks and the intentional focus on statistical validation over engineering optimizations like GNN integration. Future work will focus on extending the framework to dynamic settings. Due to space constraints, detailed discussions of limitations and social impacts are provided in Appendix Section G.

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

# A  PROOF OF DERIVATIONS $\frac{\partial \text{VEC}(\Theta)}{\partial \gamma}$:

We first consider a rank-1 tensor $\mathcal{T} = a \circ a \circ c$ with $a \in \mathbb{R}^n$ and $c \in \mathbb{R}^M$. By the definition of $\text{vec}(\mathcal{T})$, we have

$$\text{vec}(\mathcal{T}) = (c_1 (a \otimes a)^\top, \ldots, c_M (a \otimes a)^\top)^\top.$$

Denote $\{e_i\}_{i=1}^n$ as the canonical basis in $\mathbb{R}^n$. For one thing, note that

$$\frac{\partial a \otimes a}{\partial a} = \begin{bmatrix} a^\top + a_1 e_1^\top, & a_2 e_1^\top, & \ldots, & a_n e_1^\top \\ a_1 e_2^\top, & a^\top + a_2 e_2^\top, & \ldots, & a_n e_2^\top \\ \vdots, & \vdots & \vdots, & \vdots \\ a_1 e_n^\top & a_2 e_n^\top, & \ldots, & a^\top + a_n e_n^\top \end{bmatrix}^\top \tag{12}$$

$$= I_n \otimes a + a \otimes I_n, \tag{13}$$

which leads to

$$\frac{\partial \text{vec}(\mathcal{T})}{\partial a} = c^T \otimes (I_n \otimes a + a \otimes I_n). \tag{14}$$

For another, it is clear that

$$\frac{\partial \text{vec}(\mathcal{T})}{\partial c} = I_M \otimes (a \otimes a). \tag{15}$$

According to the CP decomposition of $\Theta$, we have

$$\text{vec}(\Theta) = \sum_{r=1}^R (\beta_1^{(r)} (\alpha^{(r)} \otimes \alpha^{(r)})^\top, \ldots, \beta_M^{(r)} (\alpha^{(r)} \otimes \alpha^{(r)})^\top)^\top.$$

By the property (14), we have

$$\frac{\partial \text{vec}(\Theta)}{\partial \text{vec}(\alpha)} = \begin{bmatrix} (\beta^{(1)})^\top \otimes (I_n \otimes \alpha^{(1)} + \alpha^{(1)} \otimes I_n) \\ (\beta^{(2)})^\top \otimes (I_n \otimes \alpha^{(2)} + \alpha^{(2)} \otimes I_n) \\ \vdots \\ (\beta^{(R)})^\top \otimes (I_n \otimes \alpha^{(R)} + \alpha^{(R)} \otimes I_n) \end{bmatrix}. \tag{16}$$

By the property (15), we have

$$\frac{\partial \text{vec}(\Theta)}{\partial \text{vec}(\beta)} = \begin{bmatrix} I_M \otimes (\alpha^{(1)} \otimes \alpha^{(1)}). \\ I_M \otimes (\alpha^{(2)} \otimes \alpha^{(2)}) \\ \vdots \\ I_M \otimes (\alpha^{(R)} \otimes \alpha^{(R)}) \end{bmatrix} = I_M \otimes vec(\alpha \odot \alpha). \tag{17}$$

The desired result follows from (16) and (17) immediately.

## B   FULL FORMULATION AND APPROXIMATION OF THE T-GINEE

Putting all pieces together, the T-GINEE in (3) is approximated by

$$
\sum_{i \leq j}
\begin{bmatrix}
(\beta^{(1)})^\top \otimes \Delta^{(1)} \\
\vdots \\
(\beta^{(R)})^\top \otimes \Delta^{(R)} \\
I_M \otimes (\alpha \odot \alpha)^T
\end{bmatrix}
\begin{bmatrix}
\mathrm{vec}(\mathcal{E}^{(i,j,1)})^\top \\
\vdots \\
\mathrm{vec}(\mathcal{E}^{(i,j,M)})^\top
\end{bmatrix}^\top
$$
$$
\times \left( \mathrm{diag}
\begin{bmatrix}
g'(\mathcal{P}_{i,j,1}) \\
\dots \\
g'(\mathcal{P}_{i,j,M})
\end{bmatrix}
\right)^{-1}
\widehat{\Sigma}_{i,j}(\mathcal{A}_{i,j,\cdot} - \mathcal{P}_{i,j,\cdot}(\gamma)) = \mathbf{0}, \tag{18}
$$

where $\Delta^{(r)} = I_n \otimes (\alpha^{(r)})^\top + (\alpha^{(r)})^\top \otimes I_n$ for $r \in [R]$ and $\widehat{\Sigma}_{i,j} = \Gamma_{i,j}^{1/2} \widehat{W} \Gamma_{i,j}^{1/2}$ is estimated covariance matrix.

## C   DERIVATIONS OF THEORMS

### C.1   LEMMAS

**Lemma 1.** *Under Assumptions 1-5, consider the initial estimator obtained by solving*

$$
\sum_{i \leq j} \left( \frac{\partial \mathcal{P}_{i,j,\cdot}}{\partial \gamma} \right)^\top (\mathcal{A}_{i,j,\cdot} - \mathcal{P}_{i,j,\cdot}(\gamma)) = 0, \tag{19}
$$

*using an independence working structure ($\Sigma_{i,j} = I_M$). Then, the initial estimator $\tilde{\gamma}$ is $O_p(n^{-1/2})$-consistent for the true parameter $\gamma_0$.*

*Proof.* Consider the estimating equation defined by

$$
s(\gamma) = \sum_{i \leq j} \left( \frac{\partial \mathcal{P}_{i,j,\cdot}}{\partial \gamma} \right)^\top (\mathcal{A}_{i,j,\cdot} - \mathcal{P}_{i,j,\cdot}(\gamma)) = 0.
$$

Under Assumption 1, all random variables involved, including $\mathcal{A}_{i,j,m}$, $\mathcal{P}_{i,j,m}(\gamma)$, and the derivatives of the link function $g$, are uniformly bounded for all indices $(i, j, m)$ and for all parameter vectors $\gamma$ in a neighborhood of the true parameter $\gamma_0$. Specifically, there exist constants $C_1, C_2 > 0$ such that

$$
|\mathcal{A}_{i,j,m}| \leq C_1, \quad |\mathcal{P}_{i,j,m}(\gamma)| \leq C_1, \quad \text{and} \quad |g'(\mathcal{P}_{i,j,m}(\gamma))| \geq C_2
$$

for all $(i, j, m)$ and $\gamma$ near $\gamma_0$. Additionally, Assumption 5 ensures that the partial derivatives $\frac{\partial \mathcal{P}_{i,j,\cdot}(\gamma)}{\partial \gamma}$ are bounded and that the link function $g$ is thrice continuously differentiable with uniformly bounded first and second derivatives.

By the Law of Large Numbers, as the sample size $n$ tends to infinity, the empirical sum $s(\gamma)$ converges in probability to its expectation. Specifically, at the true parameter value $\gamma_0$, the expectation of each term in the sum satisfies

$$
\mathbb{E}\left[ \left( \frac{\partial \mathcal{P}_{i,j,\cdot}}{\partial \gamma} \right)^\top (\mathcal{A}_{i,j,\cdot} - \mathcal{P}_{i,j,\cdot}(\gamma_0)) \right] = 0,
$$

since

$$
\mathbb{E}[\mathcal{A}_{i,j,\cdot}] = \mathcal{P}_{i,j,\cdot}(\gamma_0),
$$

by the model specification.

Assumption 2 guarantees that the true parameter $\gamma_0$ is uniquely identifiable as the solution to $s(\gamma) = 0$. Moreover, the low-rank condition $(n + M)R$ growing sufficiently slowly relative to $n$ ensures that the dimensionality does not impede the identification or the invertibility of the relevant Hessian matrix.

Expanding $s(\gamma)$ around $\gamma_0$ using a Taylor series, we obtain

$$
s(\tilde{\gamma}) \approx s(\gamma_0) + D(\gamma_0)(\tilde{\gamma} - \gamma_0) = 0,
$$

where $D(\gamma_0) = -\frac{\partial s(\gamma)}{\partial \gamma}\Big|_{\gamma=\gamma_0}$ is the Jacobian matrix of the estimating equations evaluated at $\gamma_0$.

From Lemma 2, we know that $D(\gamma_0)$ is invertible with eigenvalues bounded away from zero and infinity, ensuring that $D(\gamma_0)$ is well-conditioned.

Solving the linear approximation for $\tilde{\gamma}$ yields

$$\tilde{\gamma} - \gamma_0 \approx D(\gamma_0)^{-1} s(\gamma_0).$$

The term $s(\gamma_0)$ is a sum of mean-zero random variables due to

$$\mathbb{E}[s(\gamma_0)] = 0.$$

Under Assumption 4, the residuals have sub-Gaussian tails, and the central limit theorem applies, implying that

$$s(\gamma_0) = O_p(n^{1/2}).$$

Since $D(\gamma_0)$ is non-singular and its inverse has bounded operator norm, it follows that

$$\tilde{\gamma} - \gamma_0 = O_p(n^{-1/2}).$$

Therefore, the initial estimator $\tilde{\gamma}$ converges to the true parameter $\gamma_0$ at the rate of $n^{-1/2}$ in probability, establishing its $O_p(n^{-1/2})$-consistency. $\qquad\square$

**Lemma 2.** *Define $D(\gamma) = -\frac{\partial s(\gamma)}{\partial \gamma}$. Under Assumptions 1-5, the matrix $D(\gamma_0)$ is invertible with eigenvalues bounded away from zero and infinity. Furthermore,*

$$\sup_{\|\gamma-\gamma_0\|\leq 4n^{-1/2}} \|D(\gamma) - D(\gamma_0)\| = O_p(n^{1/2}).$$

*Proof.* By definition,

$$D(\gamma) = -\frac{\partial s(\gamma)}{\partial \gamma} = \sum_{i\leq j} \left[ \left( \frac{\partial^2 \mathcal{P}_{i,j,\cdot}}{\partial \gamma \partial \gamma^\top} \right) (\mathcal{A}_{i,j,\cdot} - \mathcal{P}_{i,j,\cdot}(\gamma)) + \left( \frac{\partial \mathcal{P}_{i,j,\cdot}}{\partial \gamma} \right) \left( \frac{\partial \mathcal{P}_{i,j,\cdot}(\gamma)}{\partial \gamma} \right)^\top \right].$$

Under Assumption 5, the second derivatives $\frac{\partial^2 \mathcal{P}_{i,j,\cdot}(\gamma)}{\partial \gamma \partial \gamma^\top}$ are uniformly bounded for $\gamma$ in a neighborhood of $\gamma_0$. This ensures that the first term in the expression for $D(\gamma)$ is controlled.

Assumption 2 ensures that at $\gamma = \gamma_0$, the matrix $D(\gamma_0) = \sum_{i\leq j} \left( \frac{\partial \mathcal{P}_{i,j,\cdot}}{\partial \gamma} \right) \left( \frac{\partial \mathcal{P}_{i,j,\cdot}(\gamma_0)}{\partial \gamma} \right)^\top$ is positive definite with eigenvalues bounded away from zero and infinity. This invertibility is crucial for establishing the consistency and asymptotic normality of the estimator.

To analyze the difference $D(\gamma) - D(\gamma_0)$, observe that for $\|\gamma - \gamma_0\| \leq 4n^{-1/2}$, the smoothness of $\mathcal{P}(\gamma)$ implies that

$$\left\| \frac{\partial \mathcal{P}_{i,j,\cdot}(\gamma)}{\partial \gamma} - \frac{\partial \mathcal{P}_{i,j,\cdot}(\gamma_0)}{\partial \gamma} \right\| \leq L\|\gamma - \gamma_0\| \leq 4Ln^{-1/2},$$

where $L$ is a Lipschitz constant derived from the boundedness of the first and second derivatives of $\mathcal{P}(\gamma)$ with respect to $\gamma$.

Consequently, the difference in the Jacobian matrices satisfies

$$\|D(\gamma) - D(\gamma_0)\| \leq \sum_{i\leq j} \left\| \frac{\partial \mathcal{P}_{i,j,\cdot}(\gamma)}{\partial \gamma} \frac{\partial \mathcal{P}_{i,j,\cdot}(\gamma)^\top}{\partial \gamma} - \frac{\partial \mathcal{P}_{i,j,\cdot}(\gamma_0)}{\partial \gamma} \frac{\partial \mathcal{P}_{i,j,\cdot}(\gamma_0)^\top}{\partial \gamma} \right\|.$$

Expanding the product, we get

$$\left\| \frac{\partial \mathcal{P}_{i,j,\cdot}(\gamma)}{\partial \gamma} \frac{\partial \mathcal{P}_{i,j,\cdot}(\gamma)^\top}{\partial \gamma} - \frac{\partial \mathcal{P}_{i,j,\cdot}(\gamma_0)}{\partial \gamma} \frac{\partial \mathcal{P}_{i,j,\cdot}(\gamma_0)^\top}{\partial \gamma} \right\|$$
$$\leq \left\| \frac{\partial \mathcal{P}_{i,j,\cdot}(\gamma)}{\partial \gamma} - \frac{\partial \mathcal{P}_{i,j,\cdot}(\gamma_0)}{\partial \gamma} \right\| \cdot \left\| \frac{\partial \mathcal{P}_{i,j,\cdot}(\gamma)}{\partial \gamma} \right\| + \left\| \frac{\partial \mathcal{P}_{i,j,\cdot}(\gamma_0)}{\partial \gamma} \right\| \cdot \left\| \frac{\partial \mathcal{P}_{i,j,\cdot}(\gamma)}{\partial \gamma} - \frac{\partial \mathcal{P}_{i,j,\cdot}(\gamma_0)}{\partial \gamma} \right\|.$$

Given the boundedness of the derivatives, this difference is $O(n^{-1/2})$ for each $(i, j)$. Summing over all $(i, j)$ pairs, and considering that $(n + M)R$ grows sufficiently slowly with $n$, we obtain

$$\sup_{\|\gamma - \gamma_0\| \leq 4n^{-1/2}} \|D(\gamma) - D(\gamma_0)\| = O_p(n^{1/2}).$$

Thus, the matrix $D(\gamma)$ remains close to $D(\gamma_0)$ within a neighborhood of radius $4n^{-1/2}$ around $\gamma_0$, and its deviation is controlled by $O_p(n^{1/2})$.

Furthermore, since $D(\gamma_0)$ is invertible with eigenvalues bounded away from zero and infinity, and the deviation $\|D(\gamma) - D(\gamma_0)\|$ is $O_p(n^{1/2})$, it follows that $D(\gamma)$ is also invertible in this neighborhood with high probability. The eigenvalues of $D(\gamma)$ remain bounded away from zero and infinity, ensuring the stability and invertibility of the Jacobian matrix in the vicinity of $\gamma_0$. $\square$

### C.2 PROOF OF THEOREM 3.1

*Proof.* From Lemma 1, we know that the initial estimator $\tilde{\gamma}$ satisfies

$$\|\tilde{\gamma} - \gamma_0\| = O_p(n^{-1/2}).$$

Now, consider the full estimator $\hat{\gamma}$ obtained by solving the estimating equation $s(\gamma) = 0$. Expanding $s(\hat{\gamma})$ around $\gamma_0$ via a Taylor series, we get:

$$s(\hat{\gamma}) = s(\gamma_0) + D(\gamma_0)(\hat{\gamma} - \gamma_0) + R_n,$$

where $R_n$ is a higher-order remainder term.

By Assumptions 4 and 5, we know that $s(\gamma_0) = O_p(\sqrt{n})$. Since $D(\gamma_0)$ is invertible by Lemma 2, we can solve for $\hat{\gamma} - \gamma_0$:

$$\hat{\gamma} - \gamma_0 = -D(\gamma_0)^{-1}s(\gamma_0) - D(\gamma_0)^{-1}R_n.$$

The remainder term $R_n$ is $o_p(\sqrt{n})$ due to the smoothness and boundedness conditions. Thus,

$$\|\hat{\gamma} - \gamma_0\| = O_p(n^{-1/2}),$$

establishing the consistency of the estimator. $\square$

### C.3 PROOF OF THEOREM 3.2

*Proof.* The proof follows from the central limit theorem and the consistency result established in Theorem 3.1. First, we establish a central limit result for $s(\gamma_0)$. By Assumption 4, the normalized score function $s(\gamma_0)/\sqrt{n}$ converges in distribution to a normal random variable:

$$\frac{s(\gamma_0)}{\sqrt{n}} \xrightarrow{d} N(0, M(\gamma_0)),$$

where $M(\gamma_0)$ is the variance of the score function at the true parameter $\gamma_0$.

From the consistency proof in Theorem 3.1, we know that

$$0 = s(\hat{\gamma}) = s(\gamma_0) + D(\gamma_0)(\hat{\gamma} - \gamma_0) + o_p(\sqrt{n}).$$

Rearranging this expression gives

$$\sqrt{n}(\hat{\gamma} - \gamma_0) = -D(\gamma_0)^{-1}\frac{s(\gamma_0)}{\sqrt{n}} + o_p(1).$$

Since $s(\gamma_0)/\sqrt{n} \xrightarrow{d} N(0, M(\gamma_0))$, and $D(\gamma_0)$ is an invertible matrix, applying continuous mapping arguments leads to the asymptotic distribution:

$$\sqrt{n}(\hat{\gamma} - \gamma_0) \xrightarrow{d} N\left(0, D(\gamma_0)^{-1}M(\gamma_0)[D(\gamma_0)^{-1}]^\top\right).$$

Thus, the estimator $\hat{\gamma}$ is asymptotically normal with mean $\gamma_0$ and covariance matrix $\Omega = D(\gamma_0)^{-1}M(\gamma_0)[D(\gamma_0)^{-1}]^\top$, establishing the asymptotic normality result. $\square$

## C.4 COVARIANCE ESTIMATION COROLLARY

**Corollary C.1.** *Under Assumptions 1–5, replacing $\Sigma_{i,j}^{-1}$ by $\widehat{\Sigma}_{i,j}^{-1}$ in the score function $s(\gamma)$ alters its value at $\gamma_0$ by only an $o_p(\sqrt{n})$ term. Formally, if $\widetilde{s}(\gamma)$ is defined in the same way as $s(\gamma)$ but uses $\widetilde{\Sigma}_{i,j}^{-1}$ instead of $\widehat{\Sigma}_{i,j}^{-1}$, then*

$$\|s(\gamma_0) - \widetilde{s}(\gamma_0)\| = o_p(\sqrt{n}).$$

*Proof.* By Assumption 3, we have

$$\|\widehat{\Sigma}_{i,j}^{-1} - \widetilde{\Sigma}_{i,j}^{-1}\|_F = O_p(n^{-1/2}).$$

Since all remaining factors in the construction of $s(\gamma)$ are uniformly bounded (Assumption 1) and satisfy appropriate moment conditions (Assumption 4), the difference introduced by $\widehat{\Sigma}_{i,j}^{-1}$ versus $\widetilde{\Sigma}_{i,j}^{-1}$ contributes at most $O_p(n^{-1/2})$ to each term in $s(\gamma_0)$. Summing over all $i, j$ up to order $n$ still yields an $o_p(\sqrt{n})$ overall change, since

$$(n) \times O_p(n^{-1/2}) = O_p(\sqrt{n}),$$

and thus the difference is negligible for our asymptotic analysis. $\square$

Corollary C.1 shows that using a slightly misspecified or estimated covariance in the score function $s(\gamma)$ does not affect the key asymptotic rate at $\gamma_0$. This result is crucial in ensuring that minor estimation errors in the covariance structure remain inconsequential for the consistency and asymptotic distribution of the parameter estimates. In practice, it allows us to work with convenient or empirically estimated covariance matrices without compromising the main theoretical guarantees.

## D  A FEW REMARKS FOR GINEE

First, T-GINEE is related to, but different from, generalized estimating equations (GEE, (Liang & Zeger, 1986)) or tensor generalized estimating equations (TGEE, (Zhang et al., 2019)) for generalized multivariate linear regression model with correlated predictors. Clearly, the multi-layer network $\mathcal{A}$ plays the role of response variable in GEE or TGEE. However, there is no edgewise covariate to be regressed on and the mean of $\mathcal{A}$ contains nothing but the parameters to be estimated.

Second, the rationale behind T-GINEE is that seeking a solution is a relaxation to minimize the following quadratic form

$$\frac{1}{2}\sum_{i \leq j}(\mathcal{A}_{i,j,.} - \mathcal{P}_{i,j,.}(\Theta))^{\top}\Sigma_{i,j}^{-1}(\mathcal{A}_{i,j,.} - \mathcal{P}_{i,j,.}(\Theta)). \tag{20}$$

This is because the left-hand side of (Eq. 3) is essentially the negative gradient of (Eq. 20). Herein, the precision matrix $\Sigma_{ij}^{-1}$ serves as the metric matrix (Xing et al., 2002; Liu et al., 2022) to measure the deviation of $\mathcal{A}_{i,j,.}$ to its expectation $\mathcal{P}_{i,j,.}$. When the edges $\mathcal{A}_{i,j,m}$'s, for $m \in [M]$, are independent, we have $\Sigma_{i,j} = I_M$, the $M$-th order identity matrix, and (20) reduces to the lease square loss $\frac{1}{4}\|\mathcal{A} - \mathcal{P}(\Theta)\|_F^2 - \frac{1}{4}\sum_{i=1}^{n}\|\mathcal{A}_{i,i,.} - \mathcal{P}_{i,i,.}(\Theta)\|^2$. The framework of lease square estimation for network data has been popularly employed in literature (Paul & Chen, 2020; Lei et al., 2020).

Third, a trivial solution to the GINEE (Eq. 3) as well as the minimizer of the quadratic form (Eq. 20) is $\mathcal{P} = \mathcal{A}$ if there is no further constrain in $\mathcal{P}$ or $\Theta$. This solution is meaningless and has no implication for downstream tasks of network analysis, such as network embedding, community detection, node classification, change point detection, and sub-graph density estimation. Moreover, the numbers of samples ($\mathcal{A}_{i,j,m}$'s with $i \leq j$), free parameters in $\Theta$, and unique equations in (Eq. 3) are all $n(n + 1)M/2$ due to the semi-symmetry of the multi-layer network. Thus, it is necessary to reduce the number of free parameters in $\Theta$ in order to derive a consistent estimator for $\Theta$ or $\mathcal{P}$ for subsequently tasks of multi-layer network analysis.

Fourth, selecting an appropriate rank $R$ is a crucial practical issue for tensor-based models such as T-GINEE, since it directly affects both accuracy and efficiency. Our experiments suggest a clear trade-off: higher ranks yield better accuracy but demand more computational resources. For applications

where predictive performance is paramount (e.g., biological network analysis), moderately high ranks ($R = 32$–$64$) are recommended, while in resource-constrained or real-time settings, smaller ranks ($R = 8$–$16$) provide balanced accuracy and efficiency. A practical guideline is to set $R \approx O(\log(\min(n, M)))$, where $n$ is the number.

# E  DATASETS, BASELINES AND IMPLEMENTATION DETAILS

## E.1  DATASETS

- **Krackhardt** (Krackhardt, 1987): This dataset records the cognitive social structures of a management team in a high-tech manufacturing firm, consisting of 21 managers. Each manager reported their perceived advice relationships with others, resulting in a $21 \times 21 \times 21$ tensor, where each layer corresponds to an individual's perception of the advice network.

- **AUCS** (Rossi & Magnani, 2015): The AUCS dataset consists of 61 individuals in a university setting, with five types of pairwise relations: current working, leisure activities, lunch companionship, co-authorship, and Facebook friendship. These networks form a $61 \times 61 \times 5$ multilayer adjacency tensor, facilitating the study of social group structures and community detection.

- **YSCGC** (Yeung et al., 2003): This gene co-expression dataset contains 205 genes under four functional categories, observed over 4 replicated experimental conditions. The multilayer network is constructed by thresholding pairwise gene expression similarities, resulting in a $205 \times 205 \times 4$ binary adjacency tensor for community detection and functional module discovery.

- **WAT** (De Domenico et al., 2015): The World Agricultural Trade (WAT) dataset describes trading relationships of 130 major countries across 32 agricultural products in 2010. We represent this as a $130 \times 130 \times 32$ multilayer network, with each layer indicating trade interactions for a specific product. This dataset is used for multilayer link prediction tasks.

These datasets encompass diverse network sizes and structural properties, providing a robust testbed for the effectiveness and generalizability of **T-GINEE** in multilayer network representation learning, community detection, and link prediction.

## E.2  EVALUATION METRICS:

Model performance was primarily evaluated using the Area Under the ROC Curve (AUC) on the test set, a standard metric for link prediction tasks. Additionally, we tracked both the BCE loss component (measuring prediction accuracy) and the GEE loss component (measuring correlation structure modeling) throughout training.

## E.3  IMPLEMENTATION DETAILS:

Model performance was primarily evaluated using the Area Under the ROC Curve (AUC) on the test set, a standard metric for link prediction tasks. The T-GINEE model was implemented in PyTorch, leveraging CP decomposition for efficient tensor factorization of multilayer graphs. The architecture employs node embeddings $\alpha \in \mathbb{R}^{n \times d}$ and layer embeddings $\beta \in \mathbb{R}^{m \times d}$, constructing a parameter tensor $\Theta \in \mathbb{R}^{n \times n \times m}$ through CP decomposition, which is passed through a logistic function to predict edge probabilities. After hyperparameter tuning, we selected an embedding dimension $d = 32$ for our synthetic dataset experiments, using Adam optimizer with a learning rate of $0.01$ and weight decay of $1e^{-5}$. Training proceeded with a batch size of 10,000 edges for 50 epochs, with regularization weight between BCE loss and GEE loss set to $0.1$. The working covariance matrix was updated every 5 epochs with a smoothing factor of $0.9$. We performed a grid search over embedding dimensions $d \in \{16, 32\}$, learning rates $\in \{0.001, 0.01\}$, and regularization weights $\in \{0.01, 0.1, 0.5\}$, selecting the configuration with highest validation AUC. Dataset partitioning followed an 80%/10%/10% split for training/validation/testing with a fixed random seed of 42. All experiments were conducted with PyTorch 2.2.2, with an average training time of approximately 20 minutes per dataset.

**Baselines:** To comprehensively evaluate the effectiveness of our proposed **T-GINEE** model, we compare it against a diverse set of baseline methods, encompassing classical spectral algorithms,

tensor decompositions, and matrix factorization approaches. The **Mean Adjacency Spectral Embedding (MASE)** (Han et al., 2015) approach computes the average adjacency matrix across layers and performs spectral embedding via SVD, providing a simple yet effective baseline. The **Non-negative Tucker Decomposition (NNTuck)** (Aguiar et al., 2024) method performs a non-negative Tucker tensor factorization of the multilayer adjacency tensor, optimizing factor matrices using multiplicative updates under KL-divergence loss, with three variants considering different assumptions on layer interaction. The **Spectral Kernel-based Clustering (SPECK)** (Paul & Chen, 2020) aggregates spectral information from the Laplacian matrices of all layers, constructing a consensus embedding for clustering. **HOSVD-Tucker** (Jing et al., 2021) applies higher-order singular value decomposition with Tucker decomposition to the adjacency tensor, capturing intricate multiway interactions. **Layer-wise Spectral Embedding (LSE)** (Lei et al., 2020) performs spectral clustering on each layer independently before combining results through consensus or aggregation. **CP decomposition** (Pereyra & Scherer, 1973) factorizes the adjacency tensor into a sum of rank-one tensors using alternating least-squares implementation, while **Tucker decomposition** (Tucker, 1966) generalizes CP by allowing a core tensor and separate factor matrices for each mode. **Non-negative Matrix Factorization (NMF)** (Paatero & Tapper, 1994) decomposes each adjacency matrix into non-negative factors, optionally weighting layers and applying L1 regularization, with consensus structure inferred via aggregation of factor matrices. Finally, **Singular Value Decomposition (SVD)** (Kolda & Bader, 2009) is applied to each adjacency matrix or to the mean/concatenated adjacency to extract low-rank node embeddings.

## F  Triangle Prediction on the Krackhardt Dataset

To further evaluate T-GINEE, we conducted a triangle prediction study on the Krackhardt dataset, which contains interpersonal relationship networks and is well-suited for cross-relationship prediction. We focused on triangular structures: for each triangle, one edge was removed during training and then predicted by the models. Table 3 reports accuracies across methods. T-GINEE achieves 73.36% accuracy, a 17% absolute improvement over the next best method (NNTUCK), thereby demonstrating its unique ability to leverage multilayer dependencies to infer missing relationships and validating the practical effectiveness of our theoretical framework.

Table 3: Accuracy of triangular relationship prediction on the Krackhardt dataset.

| Method | Accuracy |
|---|---|
| HOSVD | 40.33% |
| SPECK | 50.10% |
| NNTUCK | 56.42% |
| T-GINEE | **73.36%** |

## G  Limitations, Impacts and LLM Usage

### G.1  Limitations and Impacts

Although T-GINEE provides a powerful framework for tensor-based multilayer graph representation learning, several limitations still exist. The model requires sufficient network density for accurate parameter estimation, which may limit its effectiveness on extremely sparse large networks. To mitigate this, we propose augmenting sparse graphs prior to embedding by employing graph completion techniques (e.g., matrix completion, link prediction) to infer missing edges and densify the structure while preserving key patterns. In addition, our modified logit link function $g(x) = \log(x/(s-x))$ with sparsity coefficient $s$ can adaptively accommodate varying sparsity levels, allowing the model to remain effective as networks become sparser. From a scalability perspective, while CP decomposition is computationally efficient, it may not fully capture complex nonlinear dependencies, and as network size and embedding dimensions increase, computational costs grow accordingly, posing challenges for very large multilayer networks. A promising direction is to combine sparse graph representation methods (e.g., GraphSAGE sampling) with T-GINEE's tensor framework in a hybrid approach, leveraging the strengths of both. From a social impact perspective, T-GINEE can benefit multiple

domains by enhancing our understanding of complex multilayer systems, including social networks, biological interactions, and global trade relationships.

Improved representation learning can lead to more accurate predictions and better decision-making in these contexts. However, when applied to social networks, if deployed without appropriate privacy safeguards, enhanced modeling capabilities could potentially be misused for user profiling or surveillance. Additionally, like all network analysis tools, biases in the input data may be preserved or amplified in the learned representations, necessitating careful evaluation of fairness implications in sensitive applications. Finally, we emphasize that the present study does not incorporate graph neural networks or sparse-computation optimizations. This omission is intentional: our primary goal is to establish and rigorously evaluate the statistical validity and performance of T-GINEE, leaving engineering-oriented extensions to large-scale deployment as important directions for future work.

### G.2 LLM Usage Disclosure

In accordance with the ICLR 2026 policy, we disclose our use of large language models (LLMs) in preparing this manuscript. We employed OpenAI's ChatGPT and related tools primarily for language-level support, such as polishing the writing, improving grammar, and enhancing clarity across sections. The core scientific content of T-GINEE–including the theoretical development of tensor-based generalized estimating equations, formal proofs of consistency and asymptotic normality, and the design and execution of experiments on synthetic and real-world multilayer networks–was conceived, developed, and validated entirely by the authors without LLM assistance.

During manuscript preparation, LLMs were used selectively to (1) generate alternative formulations of technical explanations for readability, (2) assist with retrieval and discovery of related work and help condense lengthy descriptions in the Related Work and Conclusion sections, and (3) suggest polished phrasing for summarizing experimental results. In all cases, outputs from LLMs were carefully reviewed, validated, and substantively revised by the authors.

For implementation, we did not rely on LLMs for designing or optimizing the core T-GINEE algorithm. However, LLMs provided occasional assistance in refactoring non-core code utilities–such as dataset preprocessing scripts, figure plotting functions, and LaTeX formatting of equations and tables. All quantitative results, model derivations, and inference procedures reported in this paper remain the independent work of the authors and were verified without LLM involvement.

Overall, the role of LLMs in this project was limited to supportive tasks–text polishing, structural suggestions for presentation, and minor code formatting–while the substantive scientific contributions of T-GINEE are entirely original to the authors.

