# OpenReview forum: "T-GINEE: A Tensor-Based Multi-Graph Representation Learning"
_ICLR.cc/2026/Conference — Submitted to ICLR 2026_

### Official Review · Reviewer_XH5T · 2025-10-24

**Soundness:** 1
**Presentation:** 2
**Contribution:** 1
**Rating:** 2
**Confidence:** 5

**Summary:**

This paper introduces T-GINEE, a tensor-based generalized estimating equation framework for multilayer graph representation learning. By combining CP tensor decomposition with generalized estimating equations, the authors try to establish consistency and asymptotic normality under certain conditions. The authors also conduct experiments on synthetic and real-world networks to validate its effectiveness.

**Strengths:**

1. The authors provide some related work on network embedding and multilayer graph analysis.

2.  The authors attempt to provide a mathematical proof for their method (although it may not necessarily be correct).

 3.  The authors conduct simulation and real-world data experiments, comparing their method with existing approaches.

**Weaknesses:**

1.  The paper emphasizes the contribution of theoretical guarantees, However, the theorem and proof seem to be problematic. For instance, in Theorem 3.2, the assumption $(n + M)R = o(n^{1/3})$ is impossible, as $M,R$ are positive integers. Moreover, there appear to be errors in the proof. Some steps seem to be incorrect or not sufficiently justified.  See also question 6,7,8,9.


2. Many notations are not clearly defined, such as the notation in (1), type of norm being used in (10), and $U_{i,j,m}$ in Line 357, which makes the paper hard to follow.

 3.  The authors do not specify how to estimate Equation (3), or Equation (18) in the appendix. The estimation procedure does not seem trivial, and the authors should present an algorithm for solving these equations, along with a detailed analysis of its computational complexity.

4. The paper is the lack of rigor in the formulation of the assumptions. See my question 1, 2 and 4.

5. The numerical simulation setup is unclear, and the description of the prediction task is missing.


6. There are numerous typos throughout the manuscript. For example, in Line 259, the letter $F$ should be a subscript.

**Questions:**

1. The authors should provide a more rigorous formulation of their assumptions, rather than relying on imprecise language like "sufficiently slowly". In addition, Assumption 2 includes the statement that "The dimension $(n + M)R$ grows sufficiently slowly with $n$", but it is unclear how this condition holds, especially considering that $(n + M)R$ appears to be greater than $n$. Further clarification or a more detailed justification is needed.

2. The authors should provide a thorough discussion on the identifiability.

3. It is unclear whether $A_{i,j,\cdot}$ are independent for different pairs $(i,j)$.

4. In Line 143, the authors define $A$ as a 0-1 variable. Then the term $ (A_{i,j,\cdot} - P_{i,j,\cdot}(\Theta_0)) $ in Assumption 5 appears to be bounded, which leads to sub-Gaussian. A clearer explanation of this aspect would contribute to the rigor of the theoretical analysis.

5. The paper would benefit from additional motivation. Specifically, it is not clear why the parameter $\Theta$ is assumed to admit a tensor decomposition.  Adding more  motivation would enhance the practical contributions of the work.


6. There seems to be an issue with Line 813. In Line 806, the authors claim that for each $(i,j)$, the term (I denoted below as $F_{ij}$) is of order $n^{-1/2}$. By Line 802, we have $\|D(\gamma)-D(\gamma_0)\| = \sum_{i \le j} F_{ij}$. However, since there are at least $n$ such terms $F_{ij}$ (because $(n+M)R \ge n$), it is not clear that $\|D(\gamma)-D(\gamma_0)\|$ remains of order $n^{-1/2}$.

7. In Line 837, the author claims that $R_n = o(n^{1/2})$. This claim requires a more thorough mathematical proof, as it is not self-evident.


 8. Line 855 seems to be incorrect, as the result stated there does not follow directly from Line 852.

 9. In Line 878, the authors claim that the term is $o_p(n^{1/2})$, whereas in Line 879 the corresponding expression is $O_p(n^{1/2})$. This discrepancy should be clarified.

---

### Official Review · Reviewer_jtuF · 2025-10-31

**Soundness:** 2
**Presentation:** 3
**Contribution:** 2
**Rating:** 4
**Confidence:** 5

**Summary:**

This paper proposes a novel multilayer graph representation learning framework, T-GINEE, which combines tensor decomposition (CP decomposition) with generalized estimating equations (GEE) to explicitly capture cross-network dependencies. The core contribution of T-GINEE lies in its theoretical establishment of embedding consistency and asymptotic normality, providing robust statistical guarantees and addressing the limitations of existing methods in handling multilayer graphs. Experimental results demonstrate that T-GINEE achieves strong performance in multilayer network embedding tasks, particularly on both synthetic and real-world datasets.

**Strengths:**

The paper proposes a theoretically grounded framework for multilayer graph representation learning through careful mathematical derivation, and demonstrates its superiority in capturing higher-order structures and modeling cross-layer dependencies through comprehensive experiments on both synthetic and real-world datasets; the paper is well-structured and clearly organized, and it explicitly points out potential extensions to dynamic graph settings, providing a clear direction for future research.

**Weaknesses:**

1.	Although this paper combines CP tensor decomposition with the generalized estimating equations (GEE) framework for multilayer network representation learning and demonstrates good performance in experiments, the overall novelty of the method is limited. The core theoretical tools used in the paper—CP decomposition and GEE—are both well-established methods. The authors mainly focus on combining and engineering these existing techniques within their model, lacking substantive improvements to the theory or algorithm itself.
2.	The high computational complexity resulting from combining tensor decomposition with covariance estimation limits the scalability and practicality of the model on very large.

**Questions:**

In Figure 2(b), it can be observed that across different embedding dimensions, the AUC scores consistently increase as the regularization weight grows. Does this suggest that the regularization term plays a dominant, or even decisive, role in determining the model’s performance?

What is the advantage of the combination between CP decomposition and GEE?

---

### Official Review · Reviewer_jVAX · 2025-11-02

**Soundness:** 2
**Presentation:** 3
**Contribution:** 2
**Rating:** 4
**Confidence:** 4

**Summary:**

This paper addresses the problem of learning low-dimensional representations from multilayer graphs with incomplete node alignment and complex inter-layer dependencies. The authors propose T-GINEE that integrates CP tensor decomposition with Generalized Estimating Equations (GEE) to model structural dependencies and cross-layer correlations. The method features a flexible link function to accommodate different network sparsity structures and offers theoretical guarantees including consistency and asymptotic normality. While the proposed framework is technically sound and innovative, certain limitations remain in terms of practical applicability, scalability, and comparison depth.

**Strengths:**

1. The paper makes a substantial contribution by introducing a tensor-based GEE formulation, rigorously addressing inter-layer correlations.
2. T-GINEE significantly outperforms baseline methods across multiple benchmarks.

**Weaknesses:**

1. The paper lacks empirical evaluation on large-scale multilayer graphs (e.g., with millions of nodes). The computational cost scales linearly with embedding size (Fig. 2c), but quadratically in the number of node pairs, which can be prohibitive in practice.
2. Recent GNN-based multilayer models should be considered for fairer comparison.
3. No analysis is presented on downstream tasks such as node classification or community detection, which are important for validating representation utility.

**Questions:**

1. How does T-GINEE handle heterogeneous node sets when node alignment across layers is partial or noisy? Is any pre-alignment or matching assumed?
2. Given that the estimation uses CP decomposition, which is known to suffer from non-uniqueness, how is stability ensured in practice? Is there any regularization or initialization strategy to avoid poor local minima?

---

### Official Review · Reviewer_xP5w · 2025-11-03

**Soundness:** 3
**Presentation:** 3
**Contribution:** 3
**Rating:** 6
**Confidence:** 3

**Summary:**

This paper presents T-GINEE, a framework for representation learning on multilayer networks with partially aligned and non-identical entity correspondences. The method combines symmetric CP tensor decomposition with generalized estimating equations (GEE) to capture inter-layer correlations through working covariance matrices and flexible link functions. Theoretical guarantees on estimator consistency and asymptotic normality are provided under standard regularity conditions. Comprehensive experiments on synthetic and real-world datasets demonstrate superior predictive performance, supported by ablations, statistical analysis, and open-sourced code.

**Strengths:**

S1. A well-motivated integration of CP decomposition and GEE that rigorously models inter-layer dependencies, supported by theoretical results uncommon in graph representation learning.

S2. Strong empirical validation across multiple domains (social, biological, trade), showing consistent improvements over a wide range of baselines.

S3. The paper is clearly written, with a well-organized structure and reproducible experimental setup.

**Weaknesses:**

W1. Theoretical results rely on restrictive assumptions (boundedness, smoothness, low-rank growth) whose practical effects are not empirically tested; robustness to high-dimensional or ill-conditioned settings remains unclear.

W2. The framework is limited to (generalized) linear tensor models and does not explore integration with deep or attention-based architectures, which could enhance flexibility and expressivity.

W3. While performance metrics are strong, the experiments lack sensitivity analyses under sparsity, misalignment, and noise—key challenges for real-world multi-graph learning.

**Questions:**

Q1. How robust is T-GINEE to assumption violations such as rank mis-specification, high collinearity, or sparsity?

Q2. What are the computational or algorithmic barriers to scaling T-GINEE beyond n,M>10^3?

Q3. To what extent do the current experiments capture partial alignment rather than full alignment? Please clarify the dataset statistics or discuss practical extensions.

---

### Meta-Review · Area_Chair_MaE3 · 2025-12-30

**Summary:**

The statistical guarantees for the proposed CP+GEE multilayer embedding framework do not appear to be presented with sufficient rigor and clarity. In particular, one reviewer raises serious concerns that key theorems (like Theorem 3.2)/assumptions are implausible or contain errors and that the proof steps are not adequately justified, alongside unclear definitions. Other reviewers additionally highlight limited novelty (largely combining established works), lack of scalability evidence, and missing depth in evaluation (e.g., sensitivity to sparsity). These unresolved concerns outweigh the reported gains, leading me to recommend rejection.

**Reviewer Concerns:**

No rebuttals are provided. Thus, the reviewers' concerns won't be addressed.

**Reviewer Scores:**

No rebuttals are provided. Thus, the reviewers' scores will possibly remain the same.

---

### Decision · Program_Chairs · 2026-01-26

Reject